# Towards Streaming Synchronized Spatial Audio Generation via Autoregressive Diffusion Transformer

Ke Lei [* 1]   Yu Zhang [* 2]   Changhao Pan [* 1]   Xueyi Pu [1]   Wenxiang Guo [1]   Ruiqi Li [2]   Zhou Zhao [1]

## Abstract

Real-time and accurate spatial audio generation is pivotal for delivering an immersive experience. However, existing spatial audio synthesis technologies are often encumbered by a tradeoff between generation quality and high inference latency, as well as difficulty in capturing precise spatial information from multimodal inputs. To address these challenges, we propose SwanSphere, a unified streaming framework for high-fidelity spatial audio generation from panoramic videos and text prompts. SwanSphere mainly makes the following contributions: 1) We introduce a causal autoregressive diffusion transformer architecture that enables streaming high-quality spatial audio generation. 2) We design a Spatial Video–Audio Contrastive (SVAC) learning strategy to align the video encoder with the acoustic domain, and further employ a multi-objective online direct preference optimization (ODPO) scheme, resulting in strong spatial perception and robust multimodal spatial audio synthesis. 3) To alleviate the current scarcity of spatial audio datasets, we also develop an automated annotation pipeline for generating detailed spatial captions. Experimental results demonstrate that SwanSphere achieves superior performance in both video-to-spatial and text-to-spatial audio generation tasks. Demos can be found at: https://swansphere.github.io. Code can be found at: https://github.com/MM-Speech/SwanSphere.

## 1. Introduction

The rapid growth of Virtual Reality (VR), Augmented Reality (AR), and metaverse applications has made the construction of highly immersive audio-visual environments a central goal in multimedia research (OpenAI, 2025; Wan et al., 2025; DeepMind, 2025; Bosun, 2020). In omnidirectional video experiences, auditory immersion depends not only on high-fidelity sound reproduction but also on accurate spatial alignment between audio and visual cues (Zhu et al., 2025). To address the challenge of spatial alignment, recent work has moved beyond generating monaural audio from videos and instead focuses on video-to-stereophonic audio generation. However, jointly maintaining high quality and speed, while modeling the spatial directionality of sound sources in the ambisonic domain, remains challenging (Luo et al., 2023), making it difficult to meet the stringent spatial-consistency requirements of panoramic content.

Motivated by this gap, recent studies (Kim et al., 2025) have begun to generate First-Order Ambisonics (FOA) directly from panoramic videos, leveraging Autoregressive (AR) or Diffusion Transformer (DiT) architectures. Compared to earlier two-stage pipelines (Leng et al., 2022), several approaches achieve high-quality single-stage spatial audio generation while also improving inference efficiency (Liu et al., 2025b). Despite their promising progress, current approaches still face two key bottlenecks: 1) **Tradeoff between reconstruction errors and first-frame latency.** Mainstream regression predictions based on discrete codebooks introduce unavoidable reconstruction errors due to quantization loss (Ji et al., 2024). While large-scale Diffusion Transformer models are capable of generating high-quality audio (Wang et al., 2025), their reliance on self-attention mechanisms across global sequences and the requirement for multi-step denoising result in prolonged computation times and significantly higher first-frame latency. 2) **Hard cross-modal spatial alignment.** For most current methods, the reliance on CLIP-based encoders (Radford et al., 2021; Xu et al., 2021) results in a lack of intrinsic acoustic awareness, and global pooling tends to filter out the spatial cues necessary for accurate sound source localization (Wang et al., 2023). Taken together, the quality–latency tradeoff and the weak audio-visual spatial alignment substantially undermine the spatial coherence that is critical for truly immersive FOA synthesis.

To address these challenges, we introduce **SwanSphere**, an autoregressive diffusion framework with multimodal spatial awareness for streaming synchronized spatial audio gener-

---

*Equal contribution  [1]Zhejiang University, China [2]ByteDance, China. Correspondence to: Zhou Zhao <zhaozhou@zju.edu.cn>.

*Proceedings of the 43rd International Conference on Machine Learning*, Seoul, South Korea. PMLR 306, 2026. Copyright 2026 by the author(s).

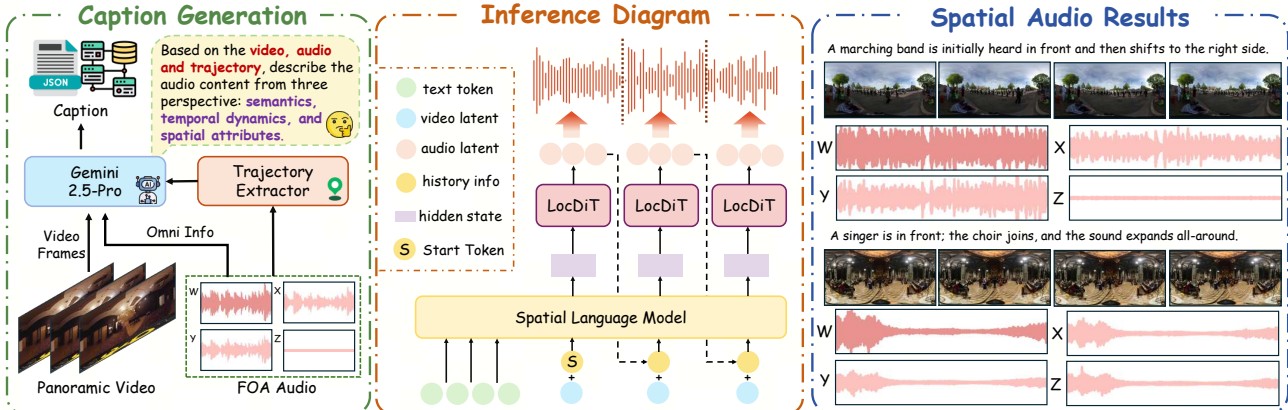

*Figure 1.* **Overview. Left**: The pipeline of audio caption generation. **Middle**: The streaming inference diagram of SwanSphere, which simultaneously supports panoramic video and textual descriptions as inputs. **Right**: Example results generated by SwanSphere. As shown above, our model accurately captures the spatial audio variation as the marching band moves from the front to the right side of the scene, manifested by a gradual decrease in audio intensity along the X-axis (front–back) and a gradual increase along the Y-axis (left–right). The example below also faithfully reproduces the immersive and enveloping sensation of a live musical performance in a concert hall.

ation. To achieve high-fidelity generation while ensuring low first-chunk latency, we introduce an autoregressive diffusion transformer framework that decouples long-range temporal modeling from local continuous rendering. In this paradigm, the autoregressive language model captures global temporal and spatial structures at the patch level, conditioned on input video tokens and captions, whereas a localized DiT (LocDiT) employs intra-patch bidirectional attention to perform denoising and synthesize high-fidelity continuous spatial audio. To better leverage spatial features in panoramic videos, we employ VideoMAE as the encoder to capture visual-spatial information and propose Spatial Video-Audio Contrastive Learning (SVAC) to enhance cross-modal spatial alignment. By designing four distinct categories of physics-aware positive and negative pairs, we align the video and audio encoders in a shared semantic space. Additionally, we implement a multi-objective online direct preference optimization (ODPO) scheme that aligns generated audio with human preferences from aesthetic, semantic, and spatial perspectives.

Furthermore, to enhance spatial perception at the data level, we devise an MLLM-based, automated annotation pipeline to curate a spatial caption-FOA dataset containing natural-language descriptions from semantic, temporal, and spatial perspectives, enabling SwanSphere to leverage both video and textual conditions. We also incorporate curriculum learning by pre-training on large-scale monaural audio, facilitating SwanSphere's adaptation to general audio distributions and achieving generalizable spatial audio generation.

As shown in Figure 1, our contributions are four-fold:

- We introduce SwanSphere, a causal diffusion transformer architecture that enables streaming high-quality spatial audio generation with multimodal inputs.

- By leveraging the SVAC strategy and multi-objective ODPO method, SwanSphere exhibits strong spatial perception capabilities from multimodal inputs, achieving alignment with human preferences across aesthetics, semantics, and spatial perception.

- To address our task requirements, we propose an MLLM-based automated annotation scheme that supports data scaling and model generalization.

- Comprehensive experiments and subjective-objective evaluations demonstrate that SwanSphere achieves strong performance across multiple dimensions, including audio generation quality and audio-visual alignment, while also exhibiting lower latency compared to baseline models.

## 2. Related Work

**Video-to-Audio Generation** Video-to-Audio (V2A) generation has recently garnered significant attention, driven by advancements in multimodal AIGC (Esser et al., 2024; Comanici et al., 2025; Siméoni et al., 2025). While prevailing V2A methodologies predominantly employ latent diffusion models (Rombach et al., 2022; Peebles & Xie, 2023), emerging research has also investigated autoregressive token-based paradigms (Touvron et al., 2023; Agostinelli et al., 2023). VTA-LDM (Xu et al., 2024) functions as a standard latent diffusion model conditioned on video features, and Diff-Foley (Luo et al., 2023) further integrates contrastive audio-visual pretraining to enhance semantic consistency. In the realm of AR generation, while SpecVQGAN (Iashin & Rahtu, 2021) utilized mel-spectrogram codebooks, subsequent works like FoleyGen (Mei et al., 2024) adopt a next-token prediction paradigm guided by visual cues. V-AURA (Viertola et al., 2025) further uses a high-frame-rate

video feature extractor to align visual features with audio tokens temporally and models audio-video co-occurrence through cross-modal feature fusion, improving temporal synchronization and semantic consistency. SoundReactor (Saito et al., 2025) extends this to a frame-level online V2A scenario, achieving end-to-end causal modeling with a causal decoder-only transformer combined with a diffusion head for online, low-latency stereo audio generation. Despite the advantages of autoregressive models in temporal alignment and online generation, these methods primarily focus on non-spatial V2A tasks (mono/stereo output) and do not explicitly model FOA spatial audio or spatial direction in panoramic video. In contrast, our work extends this line to streaming spatial audio generation from panoramic video, with additional text conditioning. Recent advancements have adopted flow matching-based generative paradigms (Lipman et al., 2022; Liu et al., 2022). Notably, MMAudio (Cheng et al., 2025) employs a flow matching framework conditioned on multimodal inputs, while Frieren (Wang et al., 2024) applies rectified flow matching with one-step distillation to enhance efficiency. To achieve superior semantic alignment, representative works such as MovieGen-Audio (Polyak et al., 2024) have further advanced the field by explicitly and jointly modeling audio, visual, and text modalities (Tang et al., 2024; You et al., 2025; Liu et al., 2025a). However, standard diffusion-based V2A models operate on global sequences, introducing significant initial latency that degrades user interactivity. Conversely, AR counterparts often lag behind in audio fidelity and perceived aesthetics (Liu et al., 2024; Huang et al., 2025). To bridge this, we propose an autoregressive diffusion transformer framework that combines textual and visual cues to achieve high-fidelity, aligned generation.

**Multimodal Spatial Audio Generation** The intrinsic spatial nature of spatial audio necessitates explicit spatial guidance during generation (Gao & Grauman, 2019). Conventional approaches typically adopt a two-stage pipeline: synthesizing mono audio followed by multimodal-guided spatialization (Morgado et al., 2018; Chen et al., 2025; Xu et al., 2021; Garg et al., 2021). However, this cascaded paradigm is heavily constrained by the quality of the initial audio generation and is prone to amplifying temporal-spatial mismatches. Recently, end-to-end methodologies have emerged, significantly improving spatial consistency (Heydari et al., 2025). For instance, BEWO (Sun et al., 2024) utilizes natural language and image inputs to synthesize binaural audio, while ISDrama (Zhang et al., 2025b) extends this by incorporating video signals and trajectory information for binaural speech synthesis. Regarding video-guided approaches, OmniAudio (Liu et al., 2025b) extracts acoustic field and semantic features from panoramic videos, whereas ViSAGe (Kim et al., 2025) generates FOA by integrating camera parameters with visual cues. Despite these advancements, prior

works predominantly leverage visual information at a semantic level, overlooking the explicit spatial cues embedded within videos. Therefore, we propose a systematic SVAC strategy, augmented by Multi-Objective ODPO to further refine spatial alignment. This approach offers a promising pathway towards more realistic spatial audio generation. Recent work has also explored controllable and stereo-aware audio generation and editing. StereoFoley (Karchkhadze et al., 2026) presents an end-to-end V2A framework for generating semantically aligned, temporally synchronized, and spatially accurate stereo audio, and further introduces a synthetic object-aware stereo pipeline that spatializes sound according to tracked object positions via dynamic panning and distance-based loudness. SmartDJ (Lan et al., 2025) investigates declarative stereo audio editing by combining an audio language model with a latent diffusion editor, where high-level user instructions are decomposed into atomic operations such as adding, removing, adjusting loudness, and relocating sound events. These studies highlight the growing importance of semantic and spatial controllability in immersive audio generation and editing. However, they mainly focus on stereo generation or editing, rather than FOA spatial audio generation from panoramic video.

## 3. Method

### 3.1. Preliminaries: First-Order Ambisonics

First-Order Ambisonics (FOA) is a specialized format for spatial audio, denoted as $\mathbf{a} \in \mathbb{R}^{C \times L}$, where $C = 4$ represents the number of channels corresponding to the $W, X, Y,$ and $Z$ components, and $L$ denotes the sampling length. In this representation, $W$ carries the omnidirectional sound pressure, while $X, Y,$ and $Z$ encode the directional velocity components along the three orthogonal axes. Our objective is to generate spatial audio $\mathbf{a}$ precisely aligned with the condition, such as visual signals or textual captions.

To map high-dimensional FOA audio into a low-dimensional latent space suitable for generative modeling, we fine-tune the established Stable Audio VAE architecture to simultaneously encode the four-channel FOA signals. Given an input signal $\mathbf{a}$, the encoder $\mathcal{E}$ maps it into a continuous latent space: $\mathbf{z} = \mathcal{E}(\mathbf{a})$, where $\mathbf{z} \in \mathbb{R}^{d \times l}$ represents the downsampled latent representation. Unlike discrete encoding methods based on codebooks (e.g., DAC), we utilize continuous latent variables for modeling. This approach effectively avoids the loss of phase information and reconstruction errors inherent in the quantization process. In this work, the audio is encoded at a frame rate of 21.5 FPS, with a latent dimension of $d = 128$ per frame. For detailed information regarding the VAE architecture and the specific training procedures, please refer to Appendix B.1.

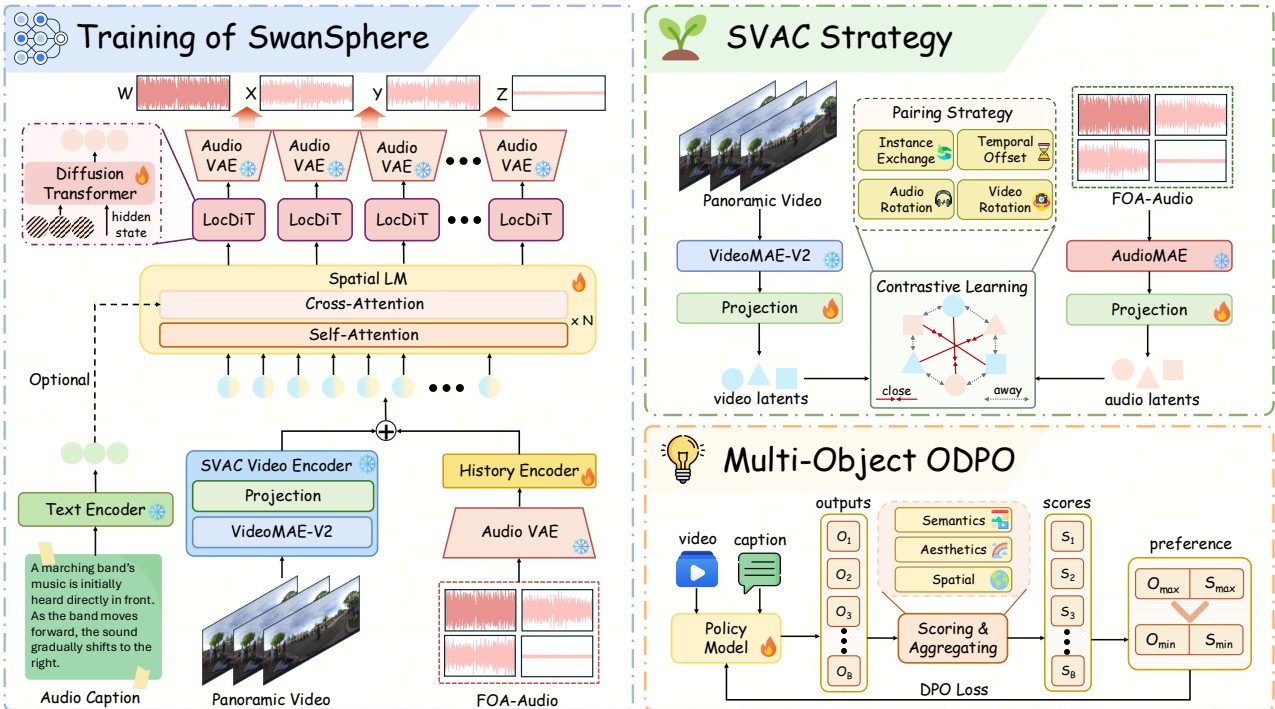

*Figure 2.* Overview of the SwanSphere framework. **The left side** illustrates the training pipeline based on the teacher forcing strategy, which supports both video and textual modalities during training. **The upper-right section** details our SVAC (Spatial Video-Audio Contrastive Learning) strategy for enhancing the Video Encoder's alignment capability. **The lower-right section** introduces the Multi-Objective Preference Alignment post-training pipeline of SwanSphere.

## 3.2. Spatial Video-Audio Contrastive Learning

Unlike traditional video-to-audio tasks, spatial audio generation requires not only semantic correspondence and temporal synchronization but also strict spatial consistency with the $360°$ visual content. Thus, we design a spatial audio-visual contrastive learning (SVAC) framework to address this challenge of panoramic-to-spatial audio generation.

For a panoramic video $v$ and FOA audio $a$, we employ a video encoder to model the input as $\mathbb{R}^{T_v \times C}$ and an audio encoder to yield $\mathbb{R}^{T_a \times C}$. Unlike prior methods that employ CLIP-based encoders for frame-wise video encoding and thus neglect geometric details, VideoMAE (Wang et al., 2023) effectively retains the video's spatial structure and temporal continuity. Since the video feature sequence has a lower temporal resolution than the audio latent sequence, we align them by nearest-neighbor replication. Specifically, each audio timestep is assigned the feature of its nearest video frame, expanding the video sequence from $T_v$ to $T_a$ without interpolating visual features. Through audio-visual contrastive learning, we inject acoustic cues from AudioMAE (Huang et al., 2022) into the visual encoder, enabling the extracted features to focus attentively on regions with acoustic potential.

We design four complementary contrastive learning objec-

tives to constrain audio-visual representations across semantic, temporal, and spatial information. (1) **Instance exchange**: at the semantic level, paired audio-visual segments constitute positive pairs, while other samples within the batch serve as negatives. (2) **Temporal offset**: to ensure accurate temporal synchronization between audio and video, we introduce temporally shifted negatives within the same video. Specifically, we apply a random circular shift to the corresponding audio to generate negative samples; this induces temporal misalignment of events, facilitating the model in learning the synchronization relationship between audio and visual onsets. (3) **Audio rotation**: given the spatial nature of FOA, we construct corresponding spatial negative samples to enhance the encoder's spatial understanding. We apply 3D rotation to the audio of the same segment to obtain spatially altered audio (with changed directional information) as negative samples. (4) **Video rotation**: for the panoramic video, we apply horizontal rotation as negative samples to reinforce the video encoder's ability to encode the geometric structure and orientation consistency of the panoramic content.

$$\mathcal{L}_{total} = \frac{1}{2N} \sum_{i=1}^{N} (\mathcal{L}_{NCE}(v_i, a_i, \mathcal{N}_i^a) + \mathcal{L}_{NCE}(a_i, v_i, \mathcal{N}_i^v))$$

$$(1)$$

To formalize these constraints within a unified framework,

we minimize the symmetric InfoNCE loss. Formally, for a batch of $N$ audio-visual pairs $\{(v_i, a_i)\}_{i=1}^N$, the total contrastive objective is defined in Eq. 1, where the unified contrastive loss $\mathcal{L}_{NCE}$ for a query $q$, a positive key $k^+$, and a set of negatives $\mathcal{M}$ is:

$$\mathcal{L}_{NCE}(q, k^+, \mathcal{M}) = \\ -\log \frac{\exp(\mathrm{s}(q, k^+)/\tau)}{\exp(\mathrm{s}(q, k^+)/\tau) + \sum_{m \in \mathcal{M}} \exp(\mathrm{s}(q, m)/\tau)}. \quad (2)$$

Here, $\mathrm{s}(\cdot)$ denotes the temporally aligned cosine similarity, and $\tau$ is the temperature parameter. Crucially, to enforce the four aforementioned objectives, we construct specific negative sets $\mathcal{N}_i^a$ (candidate audios for video $v_i$) and $\mathcal{N}_i^v$ (candidate videos for audio $a_i$) that include both batch-wise semantic negatives and spatial negatives:

$$\mathcal{N}_i^a = \underbrace{\{a_j\}_{j \neq i}}_{\text{Semantic}} \cup \underbrace{\{\tilde{a}_i^{time}\}}_{\text{Temporal}} \cup \underbrace{\{\tilde{a}_i^{spat}\}}_{\text{Spatial (Audio)}},$$
$$\mathcal{N}_i^v = \underbrace{\{v_j\}_{j \neq i}}_{\text{Semantic}} \cup \underbrace{\{\tilde{v}_i^{spat}\}}_{\text{Spatial (Video)}}, \quad (3)$$

where $\tilde{a}_i^{time}$ represents the temporally shifted audio, $\tilde{a}_i^{spat}$ denotes the spatially rotated FOA, and $\tilde{v}_i^{spat}$ is the horizontally rotated panoramic video. By discriminating the positive pair against this diverse set of hard negatives simultaneously, the video and audio encoders are compelled to learn robust representations aligned across semantic, temporal, and spatial dimensions.

### 3.3. Autoregressive Diffusion Modeling for Spatial Audio

To address the high inference latency of iterative global diffusion models and the difficulty of balancing global semantic coherence with local high-frequency details, we introduce Diffusion Transformer Autoregressive modeling for spatial audio generation (SwanSphere). Adopting a divide-and-conquer paradigm, our framework decomposes the synthesis into two stages: a semantic planning stage that produces a semantic condition $h_t$ for the current patch, and a subsequent local generation stage that synthesizes high-fidelity spatial audio conditioned on $h_t$.

The first stage is driven by a causal language model responsible for modeling inter-patch contextual dependencies. In the case of panoramic video input, at each time step $t$, the input to the model comprises the physically-aware video feature $C_v$, which is extracted via a contrastive learning module alongside the historical spatial audio information. These features, $C_v$, encapsulate both the semantic content and sound source directionality within the panoramic scene, serving as conditional guidance for the generation process. Subsequently, the LM generates a semantic embedding $h_t$

for the current time step, encoding high-dimensional semantic information to direct the subsequent generation of spatial audio latents. To ensure synchronization between the generated audio length and the video content, we utilize the termination of the video feature sequence as the stop condition, rather than relying on the model to predict a stop token. For caption inputs containing spatial information, we first extract semantic embeddings using a pre-trained FLAN-T5 (Chung et al., 2024) encoder. These embeddings are then injected into the LM via cross-attention, ensuring that the generated semantic embedding $h_t$ incorporates both the semantics and the spatial cues from the caption. To enable the model to handle different modality combinations within a unified architecture, learnable null embeddings are employed to substitute for missing modalities. Specifically, the text branch is filled with a null embedding when only video input is provided; conversely, the video features are replaced by a null embedding when only text input is available.

The second stage of generation is handled by a Local Diffusion Transformer (LocDiT), which is responsible for high-fidelity intra-patch generation. Conditioned on the LM output $h_t$, LocDiT utilizes a flow matching objective to learn the reconstruction of spatial audio latents from Gaussian noise. The history encoder summarizes the previously generated patch into a compact history representation. At step $t$, this representation is added to the aligned video tokens and then fed into the Spatial LM, enabling the LM to predict the current semantic condition with access to previous acoustic context. LocDiT receives the two preceding patches as boundary context to improve local continuity. To enhance LocDiT's capability in modeling audio latents, we also employ a curriculum learning strategy. Specifically, we convert non-spatial audio into a 4-channel format to pre-train LocDiT, equipping it with fundamental audio generation capabilities before proceeding to spatial audio training.

### 3.4. Multi-objective Online DPO

To further calibrate the generative distribution and align it more closely with the real world in terms of spatial physical laws, semantic consistency, and acoustic fidelity, we introduce a multi-objective online Direct Preference Optimization (ODPO) fine-tuning stage. Specifically, for each video or text input, the model first generates 8 candidate audio samples in parallel. These samples are then ranked via a comprehensive weighted reward function to construct preference pairs $(y_w, y_l)$. This reward function is composed of three orthogonal evaluation dimensions: First, to strictly constrain the physical accuracy of sound source localization, we utilize the errors in azimuth, elevation, and spatial angle between the generated spatial audio and the ground truth as spatial feedback. Second, leveraging the powerful cross-modal representation capabilities of ImageBind, we calculate the similarity between audio and video/text em-

beddings to ensure precise alignment of semantic content. Finally, addressing the phase distortion and mechanical artifacts commonly produced by generative models, we utilize the Audiobox Aesthetics to calculate the distance between generated audio and real reference audio in the perceptual feature space. All scores are normalized to $[0, 1]$. The total reward score is defined as the weighted sum of these three sub-objectives:

$$R = \lambda_{\text{spatial}} \cdot R_{\text{spatial}} + \lambda_{\text{semantic}} \cdot R_{\text{semantic}} + \lambda_{\text{fidelity}} \cdot R_{\text{fidelity}}, \tag{4}$$

where $\lambda_{\text{spatial}}$ and $\lambda_{\text{semantic}}$ are set to 0.4, and $\lambda_{\text{fidelity}}$ is set to 0.2. Through iterative ODPO, the model significantly eliminates neural artifacts without sacrificing generation diversity, thereby enhancing the immersion and realism of the panoramic audiovisual experience. Although the scalar reward can in principle be optimized by online RL methods such as GRPO, our sampling-and-ranking pipeline naturally yields preference pairs. We therefore adopt ODPO for stable and lightweight post-training.

### 3.5. SwanSphere Dataset Construction

To address the scarcity of high-quality spatial audio paired with aligned panoramic video and to incorporate multi-modal guidance for spatial audio generation, we curate the SwanSphere corpus through an automated data collection and annotation pipeline, enabling effective data scaling. Furthermore, we design an auxiliary scheme that utilizes non-FOA-format audio to support the curriculum learning strategy during model training.

**Data Aggregation and Preprocessing.** We aggregate raw audio from diverse open-source datasets (Liu et al., 2025b; Kim et al., 2025) and extensive web crawling to ensure comprehensive coverage, encompassing diverse physical environments (indoor and outdoor) and a wide variety of audio content, including animal sounds, natural sounds, and music. We then filter out samples with audio-visual mismatches and silent audio, and segment the data into 10-second clips. Finally, we create the SwanSphere dataset with a total of 165,000 video-audio pairs, approximately 458 hours. The test set comprises 5% of the samples, ensuring no overlapping video IDs with the training set.

**Automated spatial captioning pipeline.** To enhance spatial and temporal awareness, we design an automated annotation pipeline customized for panoramic audio-visual data. Since current Multimodal Large Language Models (MLLMs) lack the inherent capability to interpret spatial audio, directly processing First-Order Ambisonics (FOA) yields only semantic content while failing to capture accurate spatial information. To address this, we first perform sound-field analysis based on acoustic intensity vectors to estimate the azimuth, elevation, and relative distance of sound sources for each temporal segment. Temporal smoothing is then applied to

obtain continuous and stable spatial trajectories. Building on this, we feed these structured spatial trajectories, together with the panoramic video and audio, into Gemini 2.5 Pro. This process generates temporally aligned spatial audio captions that preserve physical spatial consistency. In total, we produce approximately 3,100 valid captioned samples, 300 of which are reserved for evaluation.

**Data for curriculum learning.** To enhance the generalization of SwanSphere, we construct a pre-training dataset that incorporates non-spatial audio from AudioCaps, VG-GSound, WavText5k, and AudioSet, comprising approximately 1M samples. To leverage non-spatial audio within our spatial generation framework, we adapt these signals into a pseudo-FOA format. Specifically, the omnidirectional channel $W$ is initialized as the sum of the original stereo channels. For the directional channels $X$, $Y$, and $Z$, we randomly select one channel to store the difference between the two original audio channels, while the remaining two channels are set to zero.

## 4. Experiments

### 4.1. Experimental Setup

**Evaluation metrics** To evaluate our method, we employ a dual-evaluation strategy comprising both objective measurements and subjective human studies, focusing on spatial audio fidelity and cross-modal alignment. For objective evaluation, we assess the generated audio using two categories of metrics: (1) non-spatial quality: We quantify the acoustic fidelity by calculating the Fréchet Distance (FD) between the feature distributions of the generated and reference audio, utilizing embeddings from the OpenL3 model. Furthermore, to evaluate semantic consistency, we compute the Kullback-Leibler (KL) Divergence over the label distributions predicted by a PaSST classifier pre-trained on AudioSet. (2) spatial accuracy: Following the protocols established in prior work, we evaluate the precision of sound source localization using Direction of Arrival metrics. Specifically, we report the mean absolute error for elevation ($\Delta\theta_{\text{abs}}$) and azimuth ($\Delta\phi_{\text{abs}}$), along with the aggregate angular error ($\Delta$Angular), to measure the deviation between the generated and ground-truth sound fields. To further avoid coupling the spatial reward with the evaluation protocol, we additionally adopt a pretrained SELD model, PSELD-Nets(Hu et al., 2025), as an independent spatial evaluator. For each segment and event class, PSELDNets predicts a 3D activity vector whose direction represents DoA and magnitude represents confidence. We compute the cosine similarity between generated and reference vectors and aggregate the class-wise scores weighted by their magnitudes, yielding wCS. We also include MOS-SQ (spatial audio quality) and MOS-AF (video/text-audio alignment faithfulness) for subjective evaluation.

*Table 1.* Quantitative comparison between SwanSphere and baselines. We evaluate performance across three distinct dimensions: (1) semantic quality; (2) spatial precision; and (3) efficiency. "+AS" denotes cascaded baselines utilizing an external audio spatialization. Note that for SwanSphere, inference time is reported as time-to-first-chunk / total duration.

| Model | Params | Inf. Time ↓ | FD↓ | KL↓ | $\Delta_{abs}\theta$ ↓ | $\Delta_{abs}\phi$ ↓ | $\Delta_{angular}$ ↓ | MOS-SQ ↑ | MOS-AF ↑ |
|---|---|---|---|---|---|---|---|---|---|
| Ground Truth | - | - | - | - | - | - | - | $4.60 \pm 0.15$ | $4.58 \pm 0.21$ |
| MMAudio+AS | 1.03B | 2.76s | 261.65 | 2.43 | - | - | - | $3.91 \pm 0.18$ | $3.60 \pm 0.23$ |
| Diff-Foley+AS | 0.94B | 2.03s | 304.03 | 3.12 | - | - | - | $3.68 \pm 0.14$ | $3.26 \pm 0.17$ |
| ViSAGe | 0.36B | 20.19s | 232.17 | 2.67 | 1.57 | 0.63 | 1.59 | $3.82 \pm 0.20$ | $3.78 \pm 0.26$ |
| OmniAudio | 1.22B | 0.85s | 157.67 | 1.93 | 1.25 | 0.47 | 1.27 | $4.12 \pm 0.18$ | $4.27 \pm 0.17$ |
| Ours | 1.09B | 0.21s/9.13s | **120.28** | **1.36** | **1.14** | **0.4** | **1.03** | **$4.32 \pm 0.15$** | **$4.44 \pm 0.20$** |

*Table 2.* Quantitative evaluation of generalization capabilities on text-to-spatial audio generation.

| Model | Params | Inf. Time ↓ | FD↓ | KL↓ | MOS-SQ ↑ | MOS-AF ↑ |
|---|---|---|---|---|---|---|
| Ground Truth | - | - | - | - | $4.65 \pm 0.17$ | $4.76 \pm 0.15$ |
| MMAudio+AS | 1.03B | 2.76s | 313.26 | 2.77 | $3.75 \pm 0.21$ | $3.44 \pm 0.24$ |
| AudioLDM-2+AS | 0.71B | 7.64s | 294.17 | 2.45 | $3.86 \pm 0.20$ | $3.53 \pm 0.17$ |
| Tango2+AS | 0.86B | 2.12s | 235.71 | 2.42 | $3.95 \pm 0.16$ | $3.27 \pm 0.21$ |
| OmniAudio(text) | 1.22B | 0.89s | 174.13 | 1.83 | $4.11 \pm 0.15$ | $4.16 \pm 0.18$ |
| Ours | 1.09B | 0.21s/9.13s | **142.80** | **1.43** | **$4.31 \pm 0.18$** | **$4.43 \pm 0.22$** |

**Baselines** To comprehensively evaluate our proposed method, we constructed the following baselines for comparison. For the task of generating First-Order Ambisonics (FOA) audio from panoramic video, we compare against the following methods: (1) MMAudio (Cheng et al., 2025) + audio spatialization. (2) Diff-Foley (Luo et al., 2023) + Audio Spatialization. (3) ViSAGe (Kim et al., 2025): we re-train it on our dataset using Field-of-View (FoV) video and energy maps as inputs. (4) OmniAudio (Liu et al., 2025b): As it utilizes a dual-branch architecture, we retain its original structure and re-train it on our dataset for a fair comparison. For the task of generating spatial audio from text captions, we established the following baselines: (1) MMAudio + audio spatialization. (2) Tango2 (Majumder et al., 2024) + audio spatialization. (3) OmniAudio(text): We adapted the OmniAudio architecture for text-conditional generation by incorporating caption inputs while replacing the visual inputs with learnable null tokens. We use a **signal-based spatialization strategy** based on the ground-truth audio's angles, following Ambisonics (Zotter & Frank, 2019).

### 4.2. Main Results

As shown in Table 1, the quantitative comparison results on our hybrid test set show that SwanSphere outperforms existing cascaded and end-to-end baselines across almost all objective metrics. In terms of semantic consistency, Swan-Sphere achieves an FD of 120.28 and a KL divergence of 1.36. This represents an improvement over the previous state-of-the-art OmniAudio (FD: 157.67, KL: 1.93), indicating that our generated audio aligns more closely with

the real-world distribution in both perceptual quality and semantic content. Regarding spatial accuracy, benefiting from the optimization in the ODPO stage, our model reduces the total angular error to 1.03, which is superior to OmniAudio (1.27) and ViSAGe (1.59), demonstrating more precise sound source localization capabilities. Furthermore, while maintaining a parameter count (1.09B) comparable to OmniAudio, SwanSphere achieves a first-chunk generation latency of just 0.21 seconds thanks to its streaming architecture. This demonstrates superior responsiveness for real-time interactive applications compared to the autoregressive generation of ViSAGe (20.19 seconds) and the full-sequence generation of OmniAudio. We use a patch size of 4 latent frames, temporal stride of 4, and a causal context window of 2 patches (8 latent frames). LocDiT uses 20 diffusion steps per patch. The first-chunk latency consists of 0.03s for the spatial LM, 0.14s for LocDiT denoising, and 0.04s for video encoding and audio decoding, totaling 0.21s. Larger patch sizes or longer causal context windows increase latency, highlighting the tradeoff between historical context and responsiveness. The MOS-SQ and MOS-AF metrics further demonstrate that our model achieves superior sound quality and spatial alignment in video-to-FOA generation compared to the baselines.

Beyond video-conditioned generation, we further evaluated the model's capability in the text-to-spatial audio task on our spatial audio caption test set. As shown in Table 2, we benchmarked SwanSphere against leading T2A cascaded systems (e.g., AudioLDM-2-L+AS, Tango2+AS) and OmniAudio (text), a variant adapted and retrained on our text-to-spatial audio dataset. The results demonstrate that SwanSphere

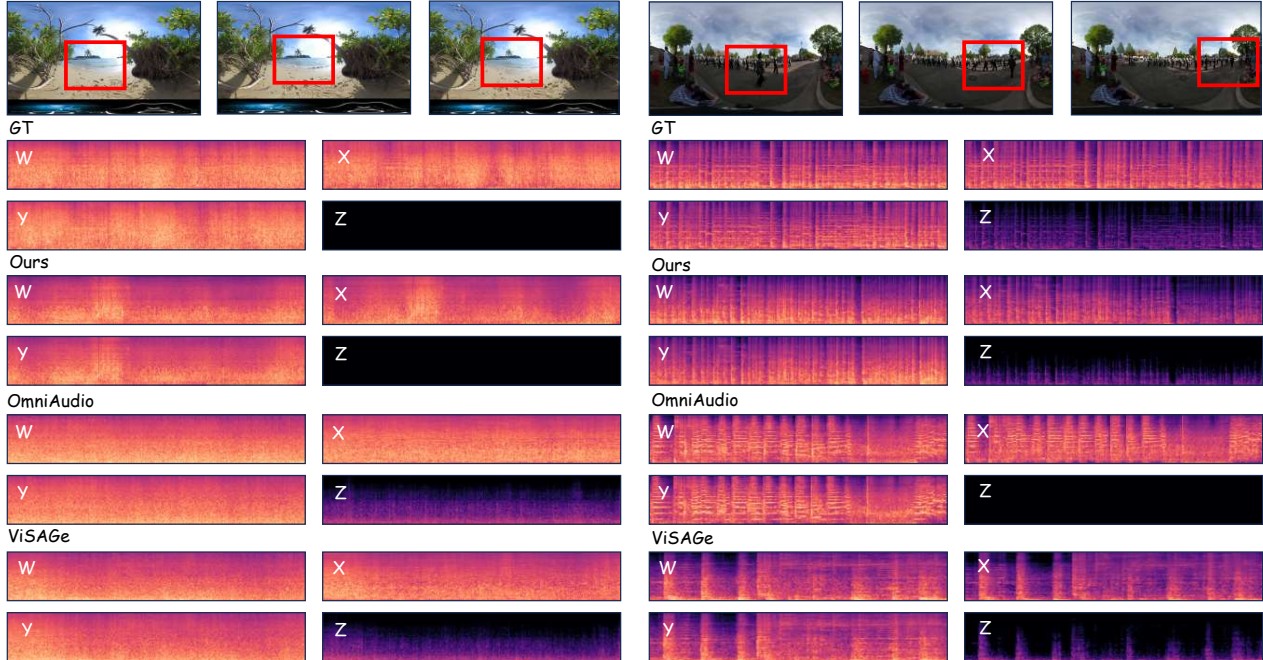

*Figure 3.* Qualitative Comparison. The left column depicts sea waves positioned directly in front; our model generates distinct and rhythmic wave sounds. In the right column, featuring a marching band moving from the front toward the right side, the signal intensity of the X channel gradually decreases while the intensity of the Y channel increases accordingly.

achieves superior performance in both semantic consistency and audio quality. Compared to the best-performing cascaded baseline Tango2+AS, our end-to-end approach exhibits a substantial lead, validating the effectiveness of unified spatial modeling. Crucially, SwanSphere outperforms the retrained OmniAudio, reducing the FD from 174.13 to 142.8 and the KL divergence from 1.83 to 1.43. These metrics indicate that our model possesses a more precise understanding of textual prompts, enabling it to generate spatial audio with higher fidelity and better alignment with real-world distributions. The two human subjective evaluation metrics further corroborate the superior sound quality of our FOA audio and its robust alignment with the provided captions. We further evaluate joint video+caption conditioning on the caption test subset. Compared with video-only conditioning, adding captions improves FD from 120.28 to 118.31 and reduces the angular error from 1.03 to 0.96, while maintaining comparable KL divergence. This suggests that captions provide explicit spatial cues complementary to panoramic video. More experimental results are shown in the appendix.

### 4.3. Ablation study

**Ablation on SVAC.** To validate the design rationale of our video feature extractor, we conducted an ablation study on the spatial video-audio contrastive learning (SVAC) strategy (as shown in Table 3). First, we established a baseline model that directly utilizes frame-level features from a pre-

*Table 3.* Ablation of spatial video-audio contrastive learning.

| Model | FD ↓ | KL ↓ | $\Delta_{abs}\theta$ ↓ | $\Delta_{abs}\phi$ ↓ | $\Delta_{angular}$ ↓ |
|---|---|---|---|---|---|
| Ours | **120.28** | **1.36** | **1.14** | **0.40** | **1.03** |
| sem-only | 127.12 | 1.41 | 1.26 | 0.49 | 1.12 |
| CLIP | 140.28 | 1.44 | 1.31 | 0.55 | 1.34 |

trained CLIP encoder as conditional input, consistent with prior research like ViSAGe (Kim et al., 2025) and Omni-Audio (Liu et al., 2025b). This configuration yielded the poorest performance (FD of 140.28, Angular Error of 1.34), indicating that generic visual encoders lack the sufficient domain adaptation capability required for fine-grained spatial audio generation tasks. Second, we evaluated the contribution of our proposed spatial-temporal negative sampling strategy. The semantic-only variant retains only semantic negative samples (i.e., mismatched audio-visual clips in a batch) while eliminating negative samples constructed based on physical laws, namely, temporal offsets and spatial rotations of FOA and panoramic videos. Compared to the full model, the exclusion of these physical negatives resulted in a marked degradation in spatial metrics (angular error increased from 1.03 to 1.12). These results compellingly demonstrate that incorporating negative samples grounded in physical laws (e.g., rotation invariance and temporal synchronization) forces the model to learn feature representations that are orientation-sensitive and temporally aligned,

Table 4. Ablation studies on SwanSphere regarding model capacity, ODPO fine-tuning, and generation paradigms.

| Model | Params | Inf. Time ↓ | FD ↓ | KL ↓ | $\Delta_{abs}\theta$ ↓ | $\Delta_{abs}\phi$ ↓ | $\Delta_{angular}$ ↓ |
|---|---|---|---|---|---|---|---|
| SwanSphere-L | 1.09B | 0.21s/9.13s | **120.28** | **1.36** | 1.14 | **0.40** | **1.03** |
| SwanSphere-M | 0.62B | 0.17s/7.60s | 132.52 | 1.43 | 1.28 | 0.46 | 1.16 |
| SwanSphere-S | 0.43B | 0.13s/6.10s | 139.81 | 1.58 | 1.45 | 0.53 | 1.33 |
| SwanSphere-L w/o ODPO | 1.09B | 0.21s/9.13s | 133.91 | 1.44 | 1.21 | 0.43 | 1.22 |
| DiT | 1.11B | 6.47s | 123.08 | 1.36 | **0.91** | 0.46 | 1.14 |

rather than merely capturing global semantic correspondences. Ultimately, the SVAC learning module significantly enhances both the spatial accuracy and auditory fidelity of the generated results. We also conduct ablation on history conditioning to verify its contribution. Specifically, we set the input to the history encoder to zero, thereby removing the history condition. The results show that FD increases from 120.28 to 128.15, and KL increases from 1.31 to 1.42; the spatial metrics also exhibit a slight degradation. These results indicate that removing historical information has an adverse effect on generation quality.

**Ablation on SwanSphere** To evaluate the impact of model capacity, generation paradigms, and the ODPO fine-tuning stage, we conducted corresponding ablation studies, with results summarized in Table 4. We investigated the necessity of sufficient model capacity by comparing the default SwanSphere-L (1.09B) against smaller variants: SwanSphere-M (0.62B) and SwanSphere-S (0.43B). As shown in the table, reducing model size leads to consistent performance degradation. Specifically, the small variant exhibits a marked decline in both semantic consistency and spatial accuracy. This indicates that the larger parameter space of SwanSphere-L is indispensable for capturing the complex correlations between panoramic vision and spatial audio, thereby validating the rationale for adopting the large architecture as our default configuration. We compared SwanSphere-L with a DiT (1.11B) version of comparable parameter size. Given that the DiT requires generating the full sequence simultaneously, it incurs an inference latency of 6.47s under 100 inference steps. In contrast, our streaming architecture achieves a Time-to-First-Chunk latency of only 0.21s, obtains better FD and aggregate angular error, and delivers an approximate 30× speedup in initial response. The ablation of the ODPO stage highlights its critical function. ODPO fine-tuning yields substantial performance gains, reducing FD from 133.91 to 120.28 and optimizing the angular error from 1.22 to 1.03. This demonstrates that ODPO successfully aligns generation with multi-objective constraints and improves the overall spatial-error profile without changing the streaming advantage of SwanSphere.

## 5. Conclusion

In this work, we present SwanSphere, a multimodal streaming framework for high-fidelity spatial audio generation from panoramic videos and text prompts. SwanSphere decouples semantic planning from local acoustic rendering via a divide-and-conquer strategy, enabling both high-quality detail reconstruction and low-latency streaming inference. To enhance spatial representation, we introduce Spatial Video-Audio Contrastive Learning and multi-objective ODPO fine-tuning, which improve orientation awareness and eliminate neural artifacts. Extensive experiments show that SwanSphere achieves state-of-the-art performance on video-to-spatial-audio and text-to-spatial-audio benchmarks, outperforming existing cascaded and unified models in semantic fidelity, spatial precision, and inference efficiency. This work advances spatial audio generation in fidelity and alignment. Moreover, its streaming architecture enables highly immersive real-time interactive environments for next-generation applications. While SwanSphere achieves state-of-the-art performance in streaming FOA spatial audio generation, it has limitations that should be acknowledged. The spatial captions primarily describe dominant sound sources, and complex multi-source scenarios (e.g., concerts with multiple simultaneous instruments) are not fully modeled, limiting fine-grained spatial disentanglement. Future work includes extending the dataset for multi-source scenes, as well as investigating robust generalization to unseen recording setups and environments.

## Impact Statement

This paper introduces a framework capable of synthesizing high-fidelity First-Order Ambisonics (FOA) that are spatially aligned with panoramic video content. While this technology holds significant promise for enhancing immersion in VR/AR, the metaverse, and multimedia content creation, it also introduces risks regarding the generation of hyper-realistic spatial deepfakes. The ability to generate spatially consistent audio-visual environments could be misused to create deceptive misinformation that is difficult to distinguish from real-world recordings. To mitigate potential misuse, we advocate for the incorporation of watermarking in the generated spatial audio and strictly restrict the open-source license to non-commercial research use. Furthermore, the spatial caption-FOA dataset constructed in this work has been rigorously screened to exclude personally identifiable information and harmful content.

## Acknowledgements

This work was supported by National Natural Science Foundation of China under Grant No.U25B2064

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

# A. Dataset Construction Details

### A.1. Video-FOA dataset construction

To build a robust and diverse foundation for spatial audio generation, we curate a large-scale composite dataset by aggregating and refining multiple sources. This includes:

- **Sphere360** (Liu et al., 2025b): 103,000 clips (approximately 288 hours) featuring 360-degree video and FOA audio.

- **YT-Ambigen** (Kim et al., 2025): 102,000 clips (approximately 142 hours) providing diverse spatial audio-visual pairs.

- Newly collected dataset from YouTube: To specifically address the long-tail distribution problem observed in existing datasets—where certain rare audio events lack sufficient representation, we collected an additional 50,000 video-FOA pairs (totaling 138 hours), following the data collection and data cleaning pipeline in (Liu et al., 2025b).

It is worth noting that the original clips in the YT-Ambigen dataset were 5 seconds long; we re-clipped them to 10 seconds to maintain consistency across the entire framework. Given the significant overlap in video content between YT-Ambigen and the Sphere360 dataset, we removed redundant samples and integrated our newly collected data. The final consolidated dataset comprises approximately 165,000 valid clips, totaling 458 hours, with 5% of the samples reserved as the test set. For all data, video was downsampled to 4 fps, and First-Order Ambisonics (FOA) audio was resampled to 44.1 kHz. Recent spatial-audio studies have also expanded recorded resources and perceptual evaluation protocols for spatial modeling (Guo et al., 2025a; Pan et al., 2025).

### A.2. Implementation details of the automated spatial captioning pipeline

Here, we provide implementation details for the automated spatial audio annotation pipeline proposed in this study. By integrating classical Digital Signal Processing (DSP) algorithms with advanced Multimodal Large Language Models (MLLMs), this pipeline addresses the deficiency of precise spatial azimuth descriptions in current datasets. The annotation pipeline comprises three core stages: acoustic spatial feature extraction, spatial trajectory smoothing, and multimodal fusion generation. Similar annotation bottlenecks also appear in neighboring audio tasks, where pitch, alignment, and style labels are often expensive to obtain (Li et al., 2024; Zhang et al., 2024c; Guo et al., 2025c).

**DoA Estimation via Acoustic Intensity Vectors** To extract sound source azimuths from First-Order Ambisonics (FOA) audio, we implemented an enhanced Direction of Arrival (DoA) estimation algorithm. We first perform a Short-Time Fourier Transform (STFT) on the four-channel audio ($W, X, Y, Z$), focusing specifically on the 500Hz - 8000Hz frequency band. This range encompasses the majority of energy for human speech and primary environmental sounds. Then we calculate the acoustic intensity vector $\vec{I} = [I_x, I_y, I_z]^T$ utilizing the cross-power spectrum between the $W$ channel (omnidirectional) and the $XYZ$ channels (pressure gradient). The calculation is as follows:

$$I_x = \text{Re}\{W^* \cdot X\}, \quad I_y = \text{Re}\{W^* \cdot Y\}, \quad I_z = \text{Re}\{W^* \cdot Z\} \tag{5}$$

To obtain a robust spatial estimate for each time segment, we aggregate these frequency-specific vectors into a unified spatial vector $\vec{V} = [V_x, V_y, V_z]^T$. This is achieved by performing an energy-weighted average of $\vec{I}$ across the selected frequency band to mitigate background noise interference. Finally, the Azimuth and Elevation are derived geometrically from the components of the aggregated vector $\vec{V}$ for each time segment (1.0 seconds):

$$\text{Azimuth} = \arctan 2(V_y, V_x), \ \text{Elevation} = \arcsin\left(\frac{V_z}{\sqrt{V_x^2 + V_y^2 + V_z^2}}\right) \tag{6}$$

**Spatial trajectory smoothing and physical consistency** To mitigate the instantaneous fluctuations inherent in direct DoA estimation and ensure physical consistency, we employ a unit vector spatial smoothing algorithm that first converts angular data into three-dimensional unit vectors $(x, y, z)$ to address the discontinuity at $\pm 180°$, followed by a moving average with a window size of 3 to eliminate computational noise. Complementing this trajectory smoothing, we estimate the relative proximity of sound sources by combining the Inverse Square Law of energy with sound field Diffuseness, providing the necessary numerical context to accurately describe dynamic movements such as *receding* or *approaching*.

**Multimodal Fusion via MLLM**   Upon acquiring the structured spatial trajectories (in JSON format), the pipeline feeds them into Gemini 2.5 Pro alongside the original panoramic video and downmixed audio.

---

**Prompt Configuration**

Role:
You are a high-precision spatial audio description generator. You must strictly adhere to the provided physical coordinate system definitions.

Coordinate System Definitions (Strict Compliance Required):

- Azimuth Definitions:

    - $0°$: Front
    - Positive Values (+): Listener's Left. (e.g., +90° is directly to the left).
    - Negative Values (-): Listener's Right. (e.g., -90° is directly to the right).
    - $\pm 180°$: Rear

- Directional Logic:

    - If the Azimuth changes from positive to negative (e.g., $+100° \rightarrow 0° \rightarrow -100°$), it indicates the object is crossing from left to right.
    - If the Azimuth changes from negative to positive (e.g., $-100° \rightarrow 0° \rightarrow +100°$), it indicates the object is crossing from right to left.

Input Data:

1. Video and Audio;

2. Spatial Trajectory (JSON): {`structured spatial trajectories here`}

Task:
Integrate the video content with the coordinate definitions above to generate a concise spatial audio description.

Output Requirements:

1. Description Elements: Sound source content, azimuth/location, approximate timing (sequence of events), and sound source dynamics (e.g., Doppler effect).

2. Style Constraints:

    - It is strictly forbidden to mention "JSON" or "algorithm" in the output.
    - Word Limit: Within 150 words.
    - Language: English.

Output Example
*A submersible emits a continuous motor hum, appearing from the front, moving rapidly to the listener's right, hovering for a moment, and then gradually fading away into the distance.*

---

# B. Implementation Details

## B.1. Latent Representation of Spatial Audio via 4-channel FOA-VAE

To effectively represent four-channel spatial audio, we initialize our spatial VAE using pre-trained weights from Stable Audio VAE (Evans et al., 2025) to leverage existing non-spatial audio knowledge. We modify the standard framework by eliminating the Mid-Side Short Time Fourier Transform (MS-STFT) originally designed for stereo reconstruction and adapt the system to transform left/right components into First-Order Ambisonics (FOA) components $W, X, Y$, and $Z$, with a loss weight of $1/4$ assigned to each channel.

As for training, we employ mixed precision with a batch size of 80 across 2 NVIDIA H800 GPUs for 200,000 steps. Subsequently, we freeze the VAE encoder and train the decoder for an additional 300,000 steps. We utilize the AdamW optimizer, setting the generator learning rate to $1 \times 10^{-5}$ and the discriminator learning rate to $2 \times 10^{-5}$. The joint optimization objective incorporates a weighted four-channel multi-resolution STFT loss, a Kullback-Leibler (KL) divergence loss applied at the VAE bottleneck, and a discrimination loss to ensure high-fidelity audio reproduction. The loss weights are

1.0, $1.0e - 5$, 1.0 respectively. We set the weight of the $L_1$ loss to zero, as empirical observations during training revealed that the $L_1$ objective exerts significant negative interference on the reconstruction of high-frequency information.

## B.2. Training Details

For the Spatial Video-Audio Contrastive Learning (SVAC) module, we first extract semantic features via the encoders and subsequently project them into a shared video-audio alignment space through projection layers (6.13M and 6.82M parameters for audio and video respectively). We initialize the video encoder with VideoMAE-V2 and the audio encoder with AudioMAE (Huang et al., 2022). Following initialization, both encoders are frozen, and only the corresponding projection layers are optimized. The training is conducted on 2 NVIDIA H800 GPUs for 100,000 steps across a dataset of 165,000 video-audio pairs, with the learning rate set to $1 \times 10^{-5}$.

Regarding the training of Swansphere, we utilize the AdamW optimizer with a learning rate of $1 \times 10^{-5}$. The model is trained for 600,000 steps on 8 NVIDIA H800 GPUs using our curated 458 hours mixed panoramic video-FOA dataset. In the subsequent multi-objective Direct Preference Optimization (DPO) stage, we conduct three rounds of online fine-tuning to further align the generative outputs with spatial and semantic preferences. Related chunkwise speech modeling has focused on low-latency conversion under streaming constraints; FOA synthesis adds the need to keep cross-channel spatial structure stable (Zhang et al., 2025d).

## B.3. Data for curriculum learning

To enhance the generalization of Swansphere, we construct a pre-training dataset that incorporates non-spatial audio from AudioCaps, VGGSound, WavText5k, and AudioSet, comprising approximately 1M samples. Broader audio generation has also made rapid progress in promptable and style-conditioned modeling (Zhang et al., 2025c; 2024b; 2025a; 2024a; Guo et al., 2025b). To leverage non-spatial audio within our spatial generation framework, we adapt these signals into a pseudo-FOA format. Specifically, the omnidirectional channel $W$ is initialized as the sum of the original stereo channels. For the directional channels $X, Y$, and $Z$, we randomly select one channel to store the difference between the two original audio channels, while the remaining two channels are set to zero.

# C. More experiments

**SELD spatial evaluator** To avoid coupling the spatial reward with the evaluation protocol, we propose using the weighted cosine similarity (wCS) metric computed by a pretrained SELD model as an independent spatial evaluator. The corresponding results are shown in the Table 5.

*Table 5.* Independent spatial evaluation using wCS metric.

| Model | MMAudio+AS | Diff-Foley+AS | ViSAGe | OmniAudio | Ours w/o ODPO | Ours |
|---|---|---|---|---|---|---|
| wCS ↑ | 0.32 | 0.27 | 0.35 | 0.41 | 0.52 | **0.63** |

**Out of distribution evaluation** To assess the generalization capability of SwanSphere beyond the training datasets, we evaluate on the YT360-Test set, which contains videos from unseen environments and microphone rigs. We compare our method with existing cascaded and end-to-end baselines in terms of both semantic quality (FD, KL) and spatial accuracy (angular metrics). As shown in Table 6, SwanSphere achieves the best performance, demonstrating robustness to out-of-distribution conditions.

*Table 6.* Out-of-distribution evaluation on YT360-Test.

| Model | Params | Inf. Time | FD ↓ | KL ↓ | $\Delta abs_\theta$ ↓ | $\Delta abs_\phi$ ↓ | $\Delta$Angular ↓ |
|---|---|---|---|---|---|---|---|
| MMAudio+AS | 1.03B | 2.76s | 276.84 | 2.64 | 1.42 | 0.57 | - |
| Diff-Foley+AS | 0.94B | 2.03s | 314.65 | 2.95 | - | - | - |
| ViSAGe | 0.36B | 20.19s | 266.91 | 2.97 | 1.82 | 0.75 | 1.86 |
| OmniAudio | 1.22B | 0.85s | 186.42 | 2.17 | 1.42 | 0.55 | 1.44 |
| Ours | 1.09B | 0.21s/9.13s | 145.83 | 1.60 | 1.24 | 0.45 | 1.13 |

