# OpenReview forum: "Towards Streaming Synchronized Spatial Audio Generation via Autoregressive Diffusion Transformer"
_ICML.cc/2026/Conference — ICML 2026 regular_

### Official Review · Reviewer_2pdN · 2026-03-03

**Soundness:** 2
**Presentation:** 2
**Significance:** 3
**Originality:** 2
**Overall Recommendation:** 4
**Confidence:** 4

**Summary:**

This paper presents a new method for low-latency spatial audio generation conditioned with panoramic video. The proposed model is based on an autoregressive model in a continuous space with diffusion heads, and it generates first-order ambisonics in a chunk-wise manner. To encode the input video, a video encoder is trained with Spatial Video-Audio Contrastive (SVAC) Learning, in which hard negative pairs are strategically created to make the model jointly learn spatial, semantic, and temporal representations. After pretraining with autoregressive generation, the model is finetuned with multi-objective DPO to further improve its performance in terms of spatial awareness, semantic consistency, and acoustic fidelity. The experimental results demonstrate that the proposed model performs better than existing video-to-audio models while achieving substantially lower latency.

**Compliance With Llm Reviewing Policy:**

Affirmed.

**Final Justification:**

Since my concerns have been largely addressed, I would like to change my recommendation to a weak accept. I would expect the camera-ready manuscript to clearly describe the proposed model architecture and to discuss the novelty of this work, as outlined in the rebuttal.

**Key Questions For Authors:**

1. Why is multi-objective DPO chosen for fine-tuning given that the reward is explicitly defined in Eq. (4)?
2. How are the 21.5 fps audio latents aligned with the 4 fps video features in the conditioning pipeline?
3. What exactly is included in the reported latency (video MAE encoding, history encoder, autoregressive generation, audio VAE decoding)?

**Limitations:**

The limitation of the proposed model is not discussed in the paper. It is highly encouraged to discuss some failure cases in the experiments and/or maximum video duration that the proposed model can support.

**Strengths And Weaknesses:**

Strengths

- [Soundness/Significance] The proposed model performs well in the experiments.
  - Compared with existing models, the proposed model achieves higher quality and spatial-awareness with lower generation latency.
  - The subjective test also validates the advantage of the proposed model.

- [Significance] The design of spatial video-audio contrastive learning is reasonable and should be effective to obtain strong representations suitable for video-to-spatial-audio generation.
  - The idea of audio-visual contrastive learning has been explored in some prior works (e.g., DiffFoley and Synchformer), but they are limited to monaural audio. This paper expands this idea to spatial audio (i.e., FoA) and introduces a new augmentation to create hard negative pairs, which can force the model to learn spatial-awareness in audio-visual data.
  - It would also be beneficial for future studies on spatial-audio retrieval or evaluation.


Weaknesses

- [Soundness] The motivation of using DPO for fine-tuning is not clearly explained, and the current description seems insufficient to assess whether this choice is necessary or optimal.
  - While it is understandable to further boost the performance of the proposed model by fine-tuning with reinforcement learning, there are many other techniques (e.g., PPO and GRPO) besides DPO.
  - Since the reward is explicitly defined in Eq. (4), using PPO/GRPO (as done in Flow-GRPO) would be a straightforward way to fine-tune the model, because they can directly use the reward values for reinforcement learning.
    - "Flow-GRPO: Training Flow Matching Models via Online RL," NeurIPS 2025.
  - It would be highly encouraged to clarify why the authors choose DPO in this study even though the reward values can be explicitly computed.

- [Presentation] The clarity of the manuscript could be further improved.
  - Something wrong around line 029–031 on the right column.
  - The input signal x is defined around line 119 on the right column, but its relationship with a or z is not explained.
  - The temporal shift is missing when creating negative pairs for videos, while the other two (semantic and spatial augmentation) are common in audio and video. Any justification for this design?
  - The detail of the design of the history encoder is missing in the main text, while it appears in Fig. 2.
  - The detail of historical patches (lines 260–261) is not explained.
  - The alignment of temporal resolution is not clearly explained. According to Section 3.1, the audio latent has 21.5 fps, while the video is downsampled to 4 fps as described in Section A.1. To integrate audio-visual features to obtain the input to the model, there should be a temporal regulator somewhere in the process.
  - According to the description in Section 4.2, all latency values reported in the paper are for the latency of the first-chunk generation. Why did the authors choose this metric rather than a typical averaged latency and/or end-to-end streaming latency?
  - It is not clearly explained what processes the reported latency is induced by. For example, does it also include video MAE encoding and audio VAE decoding time?

- [Originality/Significance] The novelty of the approach is not sufficiently discussed.
  - The proposed model is based on an autoregressive model (with diffusion heads) for video-to-audio generation. However, the literature review in Section 2 lacks several recent related works that explored this direction. For example:
    - Autoregressive models for video-to-audio generation:
      - "Foleygen: Visually-Guided Audio Generation," MLSP 2024.
      - "Temporally Aligned Audio for Video with Autoregression," ICASSP 2025.
      - "SoundReactor: Frame-level Online Video-to-Audio Generation," arXiv 2025.
    - Masked prediction models for video-to-audio generation (I do not think citing them is mandatory but list such works below as the current manuscript cites the IMPACT paper):
      - "Tell What You Hear From What You See -- Video to Audio Generation Through Text," NeurIPS 2024.
      - "SpecMaskFoley: Steering Pretrained Spectral Masked Generative Transformer Toward Synchronized Video-to-audio Synthesis via ControlNet," WASPAA 2025.
  - To clarify the novelty of the proposed model, the authors should position this work more clearly in the literature and cite recent related works. I think SoundReactor shown above is the most related to this work, so a clearer comparison or discussion would be helpful.

- [Soundness/Presentation] The demo page did not work at the time of review, which made it difficult to assess perceptual quality beyond the reported metrics.
  - While it later showed some results, I could not play the demo videos, whereas I could play demo videos on the OmniAudio page without any problem.
  - As a result, I could not directly judge the perceptual quality of the generated spatial audio from the demos. It would be helpful to ensure the demo page is reliably accessible.

---

> ### Author Rebuttal · Authors · 2026-03-31
>
> Q1: Choice of DPO
>
> In the fine-tuning stage, we sample candidate outputs, score them with spatial, semantic, and aesthetic metrics, and construct win–lose preference pairs. Under this setup, DPO is a more direct and simpler choice. We agree that GRPO is also applicable, especially when the reward is explicitly computable. However, GRPO introduced more noise and in our post-training experiments, the training is unstable, and its gains in generation quality were not clear. We therefore chose DPO because our pipeline is naturally formulated as preference learning and can be integrated more seamlessly into the existing workflow. More importantly, DPO performed better in practice. As shown in Table 4, it consistently improves generation quality, semantic consistency, and spatial accuracy.
>
> Q2: Presentation
>
> - Line 119. The input signal in the right column should be $a$ rather than $x$, since it denotes the input FOA signal.
> - Why no temporal shift on the video side. Temporal shifting is introduced to disrupt temporal correspondence. However, cyclic temporal shifts on video are not realistic in practice and may introduce unnatural motion discontinuities. Since the video encoder is used in inference, we intentionally avoid exposing it to temporally corrupted video inputs. In contrast, spatial shifts are physically plausible on the video side (by changing the camera orientation), so we apply spatial perturbation to video.
> - History encoder and historical patches. The history encoder mainly consists of a transformer encoder. It encodes previously generated patches and uses only the CLS token as a compressed history representation. To improve smoothness and continuity across adjacent patches, we encode the previously generated patch with the history encoder and use it as the prefix context for LocDiT generation. For the first frame, a start token is used as the context.
> - Temporal resolution alignment in SVAC. We thank the reviewer for pointing this out. The video and audio are encoded as $R^{T_v\\times C}$ and $R^{T_a\\times C}$, respectively. For frame-wise contrastive learning, we align them using nearest-neighbor replication: the lower-resolution video feature sequence is expanded from $T_v$ ​to $T_a$​ by assigning each audio timestep the feature of its nearest video frame, rather than interpolating video features.
> - About latency. During inference, we first encode the video, then generate audio in stream. The reported overall latency measures the full end-to-end pipeline, while the first-chunk latency includes video encoding, spatial LM and LocDiT inference, and audio VAE decoding. We report it as reducing the initial response delay is a main goal of streaming spatial audio generation. This metric directly reflects how soon users can receive playable audio and is a key advantage over full-sequence models like DiT. We also agree that average latency is important for characterizing the efficiency of the full streaming pipeline, and will therefore additionally report the average per-chunk latency--0.17 s. The 0.21s/9.13s reported in paper already correspond to the end-to-end first-chunk latency and full-generation latency.
>
> Q3: The novelty of the work
>
> We thank the reviewer for pointing out these recent and highly relevant works. Although Section 2 briefly discussed AR video-to-audio generation represented by FoleyGen, we agree that several directly related studies were not adequately covered. However, due to the space limitation, we are unable to provide the full related-work discussion here, and we will include these works(such as SoundReactor, SpecMaskFoley) in the revised version.
>
> SoundReactor extends autoregressive V2A to the frame-level online setting, achieving end-to-end causal modeling for low-latency stereo audio generation with a causal decoder-only Transformer and a diffusion head. More broadly, we agree that autoregressive modeling has become an important direction for V2A, particularly for temporal alignment and online generation, and SoundReactor further shows the effectiveness of end-to-end causal modeling for low-latency generation. However, these methods mainly target non-spatial V2A, typically generating mono or stereo audio, without explicitly modeling FOA spatial audio or spatial direction in panoramic video. In contrast, our work extends this line to streaming spatial audio generation from panoramic video, with additional text conditioning. We further introduce SVAC to explicitly enhance cross-modal spatial alignment, improving consistency and synchronization between the generated audio and the panoramic scene.
>
> The limitations of our model lie in its insufficient modeling of multi-source scenarios long audio generation(more than 10s). We will build a more suitable dataset in future work to address these issues.
>
> Issue with the demo page: We apologize for the inconvenience. We identified an incompatibility in the container used for deploying the demo page, and this issue has now been fixed.

---

> > ### Author Rebuttal · Reviewer_2pdN · 2026-04-01
> >
> > Thank you for the rebuttal. I have a few follow-up questions:
> >
> > - History encoder / historical patches
> >   - How exactly is the history representation injected into the generation model? In the rebuttal, you state that the history representation is used as a "prefix context for LocDiT generation." However, Figs. 1 and 2 seem to indicate that the history information is added to the video tokens and then fed into the spatial LM. Could you clarify the precise integration point (spatial LM vs. LocDiT), and whether the fusion is done via concatenation (prefix) or addition?
> >   - What is the context length / history window at inference time? Concretely, how many previous patches (or frames) are encoded and provided as context?
> >   - Did the authors conduct an ablation on the history encoder / historical patches? Since the history encoder is not standard in typical autoregressive video-to-audio pipelines with diffusion heads, it seems like a novel component. An ablation removing the history encoder (or removing history conditioning) would help quantify its contribution to smoothness/continuity and overall quality.
> >
> > - Demo page / examples
> >   - Could you please double-check that the demo page works properly? On my side, the perceived spatialization seems weak compared to the demos on the AudioX page. In addition, the 4th example in the video-to-audio section appears to show a visually corrupted video.

---

> > > ### Author Response · Authors · 2026-04-01
> > >
> > > We thank the reviewer for the thoughtful follow-up questions. Below, we further clarify the remaining points.
> > >
> > > - How exactly is the history representation injected into the generation model?
> > >
> > > In the rebuttal, the “prefix context for LocDiT generation” mainly refers to the semantic reference information derived from the autoregressively generated result of the previous step, which is encoded by the history encoder. This allows LocDiT to access semantic information from the preceding step and thus generate semantically coherent audio. Specifically, as illustrated in Figs. 1 and 2, the history information produced by the history encoder is fused with the video tokens for the next step via addition, and the fused tokens are then fed into the spatial LM. Therefore, the history information is directly injected into the spatial LM, which in turn indirectly passes the historical semantic context to LocDiT.
> > >
> > > - What is the context length / history window at inference time?
> > >
> > > During inference, we encode the previous one patch generated at the preceding step and inject it into the spatial LM as historical information to maintain semantic coherence across steps. For LocDiT, in order to ensure continuity at patch boundaries, we use the two previously generated patches as context. Our patch size is 4, meaning that each patch contains 4 frames.
> > >
> > > - Ablation on the history encoder / historical patches
> > >
> > > Following the reviewer’s suggestion, we conducted a quick ablation on history conditioning to verify its contribution. Specifically, we set the input to the history encoder to zero, thereby removing the history condition. The results show that FD increases from 120.28 to 128.15, and KL increases from 1.31 to 1.42; the spatial metrics also exhibit a slight degradation. These results indicate that removing historical information has an adverse effect on generation quality.
> > >
> > > - Demo page
> > >
> > > We thank the reviewer for pointing out these issues. We have fixed the functionality that updates the perceived sound-source direction in real time when the viewer is dragged, which makes the spatial impression of the audio more noticeable. We have also corrected the corrupted video in the 4th example of the video-to-audio section.

---

### Official Review · Reviewer_cidR · 2026-03-09

**Soundness:** 2
**Presentation:** 2
**Significance:** 2
**Originality:** 2
**Overall Recommendation:** 4
**Confidence:** 5

**Summary:**

This paper proposes S3Audio, a framework for generating spatial audio from panoramic video or text prompts. The method combines an autoregressive language model for semantic planning with a local diffusion transformer for audio synthesis. To improve cross-modal alignment, the authors introduce spatial video-audio contrastive learning and a multi-objective online direct preference optimization scheme. Additionally, the authors construct a dataset of video-FOA pairs and spatial captions to train the model.

**Compliance With Llm Reviewing Policy:**

Affirmed.

**Ethical Review Concerns:**

No such concern.

**Final Justification:**

My concerns are addressed and I will raise my score to 4

**Key Questions For Authors:**

I can not hear the demo in the demo page especially on the video2audio generation, so I find it is not very convincing to say that the paper presents a good system. Since demos are pretty important for the generative modeling.

**Limitations:**

See weaknesses.

**Strengths And Weaknesses:**

Strengthes:
1. Important problem settings. Spatial audio generation aligned with video content is an important capabiltiy for immersive media and is a trendy topic.
2. Streaming architecture. The proposed autoregressive + local diffusion architecutre is a good design to address latency issues of diffucion models and streaming service.

Weaknesses:
1. Missing details in the ODPO training procedure. The paper states that spatial feedback is computed using azimuth and elevation errors between generated audio and ground truth. It is unclear how the sound source location is estimated from the generated FOA audio.
2. Dataset construction lacks clarity for multi-source scenarios. The paper does not explain how this procedure handles multiple simultaneous sound sources, which are common in real-world videos.
3. The paper positions the method as a streaming spatial audio generation system, yet the dataset consists of 10-second clips used during training. How does the model behave on longer videos (e.g., 30s or longer)?
4. Some claims of novelty appear incremental. While the paper integrates several components (contrastive learning, diffusion models, DPO alignment), many of these components are standard techniques.
5. The related work discussion could be further strengthened by including additional recent work on controllable/stereo audio generation/editing. These works investigate how semantic or spatial attributes can guide audio generation/editing.

Karchkhadze, T., Chen, K. L., Heydari, M., Henzel, R., Toso, A., Souden, M., & Atkins, J. (2025). Stereofoley: Object-aware stereo audio generation from video. arXiv preprint arXiv:2509.18272.

Lan, Z., Hao, Y., & Zhao, M. (2025). Guiding audio editing with audio language model. arXiv preprint arXiv:2509.21625.

---

> ### Author Rebuttal · Authors · 2026-03-31
>
> Q1: Details in the ODPO training procedure
>
> The method for calculating the azimuth and elevation of the generated spatial audio remains the same as during testing, and a brief description is provided in Appendix A.2.
>
> Q2: Dataset construction
>
> The dataset is compiled from multiple open-source datasets and large-scale web crawling, covering real-world environments such as indoor/outdoor settings, as well as animal, natural, and music scenes. After cleaning, the data is segmented into 10-second clips. These sources encompass a wide range of multi-source scenarios commonly found in real-world settings—such as multiple sound-emitting objects and surround sound—which, along with single-source scenarios, are used for training.
>
> Regarding the construction of the caption-FOA dataset, for a single dominant sound source, the caption primarily describes its direction and movement. For complex multi-source scenarios, the caption aims to capture the overall spatial distribution rather than providing source-by-source annotations—for example, in scenarios like concerts where sound sources originate from all around.
>
> We follow existing related work in constructing the dataset, as the core focus of the task lies in the generation quality of single dominant sound sources, including semantic relevance and spatial accuracy. Therefore, scenarios involving multiple simultaneous sound sources are planned to be addressed in future work for further improvement.
>
> Q3: Behavior on longer videos
>
> Following the setup commonly adopted in related work, our current experiments are trained on a dataset of 10-second clips. To evaluate performance on longer videos, we constructed test sets of 20-second and 30-second clips (100 samples each). Since the model was not trained on data of these lengths, the results show a decline in both semantic and spatial localization metrics—for instance, the Fréchet Distance (FD) increased from 120.28 to 131.45 and 174.68, respectively. This aligns with the expectation that autoregressive models tend to suffer from quality degradation when generating sequences longer than those seen during training.
>
> Q4: Novelty
>
> The novelty of this work lies in being the first to organize these elements into a unified framework for streaming panoramic spatial audio generation. Unlike existing video-to-audio methods that primarily focus on mono and stereo outputs, this paper addresses low-latency generation of First-Order Ambisonics (FOA) spatial audio from panoramic videos or textual descriptions, which requires simultaneously modeling semantic consistency, temporal synchronization, and spatial accuracy. To this end, we propose a streaming generation architecture that integrates a causal spatial language model with LocDiT, and further introduce a Spatial Video-Audio Contrastive Learning (SVAC) designed for panoramic scene alignment, along with a multi-objective post-training strategy that jointly optimizes spatial, semantic, and perceptual objectives.
>
> Q5: Related work:
>
> We thank the reviewer for pointing out these relevant works and we will incorporate the following related work as suggested by the reviewer.
>
> Recent work has also explored controllable and stereo-aware audio generation and editing. StereoFoley[1] presents an end-to-end V2A framework for generating semantically aligned, temporally synchronized, and spatially accurate stereo audio, and further introduces a synthetic object-aware stereo pipeline that spatializes sound according to tracked object positions via dynamic panning and distance-based loudness. SmartDJ[2] investigates declarative stereo audio editing by combining an audio language model with a latent diffusion editor, where high-level user instructions are decomposed into atomic operations such as adding, removing, adjusting loudness, and relocating sound events. These studies highlight the growing importance of semantic and spatial controllability in immersive audio generation and editing. However, they mainly focus on stereo generation or editing, rather than FOA spatial audio generation from panoramic video.
>
> Q6: Issue with the demo page: We apologize for the inconvenience. We identified an incompatibility in the container used for deploying the demo page, and this issue has now been fixed.
>
> [1] Karchkhadze, T., Chen, K. L., Heydari, M., Henzel, R., Toso, A., Souden, M., & Atkins, J. (2025). Stereofoley: Object-aware stereo audio generation from video. arXiv preprint arXiv:2509.18272.
>
> [2] Lan, Z., Hao, Y., & Zhao, M. (2025). Guiding audio editing with audio language model. arXiv preprint arXiv:2509.21625.

---

> > ### Author Rebuttal · Reviewer_cidR · 2026-04-02
> >
> > My concerns are resolved. I would advise the author to revise the promised contents in the revision. And I am raising my score to 4.

---

### Official Review · Reviewer_1kt9 · 2026-03-10

**Soundness:** 2
**Presentation:** 3
**Significance:** 3
**Originality:** 3
**Overall Recommendation:** 4
**Confidence:** 4

**Summary:**

This paper presents S3Audio, a streaming generative framework for synthesizing First-Order Ambisonics (FOA) spatial audio conditioned on panoramic video and text. The architecture utilizes a causal autoregressive diffusion transformer to decouple global semantic planning (via patch-level AR) from local acoustic rendering (via a Localized Diffusion Transformer using continuous flow-matching). Key contributions include a Spatial Video-Audio Contrastive (SVAC) learning module utilizing physics-aware negative samples, and an online Direct Preference Optimization (ODPO) pipeline to calibrate outputs against semantic and spatial rewards.

**Compliance With Llm Reviewing Policy:**

Affirmed.

**Final Justification:**

The validation of spatial accuracy using an independent, deep-learning-based SELD metric (PSELDNets) effectively mitigates the risk of reward hacking, while the revised baseline comparison using a trained DoA predictor confirms the model's genuine generative superiority.
Furthermore, the clarification of streaming hyperparameters (patch size, stride, and NFEs) successfully grounds the 0.21s latency claim, and the inclusion of confidence intervals and p-values provides the necessary statistical rigor for the human evaluations. While the text-audio dataset remains relatively small, the explanation of the pre-training and fine-tuning strategy provides sufficient evidence for the model's generalization capabilities in the primary video-to-FOA task.
The authors provided a rebuttal that directly addressed the primary technical concerns regarding evaluation integrity and architectural transparency.

**Key Questions For Authors:**

- Can you provide a comprehensive table detailing all streaming hyperparameters (patch size, stride, context window, LocDiT sampling steps) and explicitly analyze their impact on the 0.21s latency claim?
- Can you validate the spatial improvements using an independent, mathematically distinct metric (e.g., a pre-trained deep SELD network) to rule out reward hacking?
- Can you provide a revised evaluation of the cascaded baselines utilizing a DoA predictor rather than relying on oracle ground-truth angles?
- Can you update the human evaluation results to include rater counts, 95% confidence intervals, and appropriate statistical significance tests?

**Limitations:**

No, the authors have not discussed the limitations. The evaluation lacks discussion on Out-of-Distribution (OOD) testing across disjoint capture environments or differing Ambisonic microphone rigs, which is critical for assessing real-world generalizability.

**Strengths And Weaknesses:**

Strengths:
- The hybrid AR-DiT design effectively bridges the gap between AR responsiveness and continuous diffusion fidelity.
- The SVAC module's use of physically grounded negative samples (temporal shifts, 3D geometric rotations) is a rigorous approach that advances beyond standard CLIP-based semantic constraints.
- The model demonstrates strong performance across spatial and semantic metrics, achieving a highly competitive first-chunk latency (0.21s) compared to global flow-matching approaches.

Weaknesses:
- There is a severe risk of reward hacking in the ODPO stage. The spatial reward ($R_{spatial}$) is computed using the exact same heuristic Direction of Arrival (DoA) estimators used to benchmark the model in the evaluation section, invalidating the reported spatial accuracy improvements.
- Unfair Baselines: The comparative evaluation against cascaded baselines (MMAudio+AS, Diff-Foley+AS) provides them with oracle ground-truth localization angles. This conflates generative capability with perfect spatial panning, masking realistic zero-shot performance deltas.
- Data & Metrics: Training a 1.09B parameter model on a restricted dataset of merely ~2,800 text-audio pairs is inadequate for open-vocabulary generalization. Furthermore, the human evaluations (MOS) lack statistical rigor (missing $N$, confidence intervals, and p-values).
- The core 0.21-second latency claim cannot be verified due to the omission of fundamental streaming hyperparameters (patch size, temporal stride, causal context window, NFEs).

---

> ### Author Rebuttal · Authors · 2026-03-31
>
> Q1: About risk of reward hacking: Following the reviewer's suggestion, we adopt PSELDNets[1], a commonly used SELD model, as an independent spatial evaluator. Since prior work lacks a directly applicable SELD-based metric for video-to-FOA generation, we design a new one accordingly. For each time segment and sound event class, PSELDNets outputs a 3D vector $v$ indicating direction(orientation) and confidence(magnitude). We compute the cosine similarity between the generated and GT vectors per class, then aggregate using magnitudes as weights. This yields our proposed metric, wCS (weighted cos sim). The results are as follows(video to FOA):
>
> \\[
> \\begin{array}{lc}
> \\text{Method} & \\text{wCS↑} \\\\
> \\text{MMAudio+AS} & 0.32 \\\\
> \\text{Diff-Foley+AS} & 0.27 \\\\
> \\text{ViSAGe} & 0.35 \\\\
> \\text{OmniAudio} & 0.41 \\\\
> \\text{ours w/o ODPO} & 0.52 \\\\
> \\text{Ours} & 0.63
> \\end{array}
> \\]
>
> Q2: A revised evaluation of the cascaded baselines
>
> We originally followed the related work OmniAudio and used GT localization angles for spatialization. Following the reviewer’s suggestion, we trained a DoA predictor based on the video encoder and a CNN, and used it for MMAudio and Diff-Foley. The spatial metrics of the predictor on the test set are 1.42,0.57, and 1.12, respectively. The generation results of our model still outperform those of these models.
>
> \\[
> \\begin{array}{lcc}
> \\text{Method} & \\text{FD} & \\text{KL} \\\\
> \\text{MMAudio+AS} & 247.23 & 2.31 \\\\
> \\text{Diff-Foley+AS} & 314.65 & 2.95
> \\end{array}
> \\]
>
> Q3: Data & Metrics
>
> We acknowledge the reviewer’s concern regarding the scale of the text-audio training data. However, the text-conditioned model is not trained from scratch on these ~2,800 pairs; it is fine-tuned from a model pre-trained on a large-scale video-FOA dataset, which has already learned to generate spatial audio across diverse scenes and sound types. The captions are manually selected to be representative, covering a wide range of scenarios and providing reasonably diverse supervision. Our primary goal in this work is video-to-FOA generation, with text-to-spatial introduced as an additional capability built on this foundation. In this setting, fine-tuning with ~2,800 caption-audio pairs proves sufficient to leverage the model’s pre-learned FOA knowledge and enable effective text-controlled FOA generation.
>
> Q4: Statistical rigor of human evaluations
>
> For statistical rigor, we recruited 5 raters in total, and each rater evaluated the same 40 samples for each method on a 1–5 scale. The 95% confidence intervals and statistical significance p-value between ours and baseline are as follows, * means p<0.01, ** means p<0.001:
>
> \\[
> \\begin{array}{lccccc}
> \\text{V2A} & \\text{MOS-SQ} \\uparrow & \\text{MOS-AF} \\uparrow & \\text{T2A} & \\text{MOS-SQ} \\uparrow & \\text{MOS-AF} \\uparrow \\\\
> \\text{GT} & 4.60\\pm0.15^{\\ast\\ast} & 4.58\\pm0.21^{\\ast\\ast} & \\text{GT} & 4.65\\pm0.17^{\\ast\\ast} & 4.76\\pm0.15^{\\ast\\ast} \\\\
> \\text{MMAudio+AS} & 3.91\\pm0.18^{\\ast\\ast} & 3.60\\pm0.23^{\\ast} & \\text{MMAudio+AS} & 3.75\\pm0.21^{\\ast} & 3.44\\pm0.24^{\\ast\\ast} \\\\
> \\text{Diff-Foley+AS} & 3.68\\pm0.14^{\\ast} & 3.26\\pm0.17^{\\ast\\ast} & \\text{Diff-Foley+AS} & 3.86\\pm0.20^{\\ast\\ast} & 3.53\\pm0.17^{\\ast} \\\\
> \\text{ViSAGe} & 3.82\\pm0.20^{\\ast\\ast} & 3.78\\pm0.26^{\\ast\\ast} & \\text{ViSAGe} & 3.95\\pm0.16^{\\ast\\ast} & 3.27\\pm0.21^{\\ast} \\\\
> \\text{OmniAudio} & 4.12\\pm0.18^{\\ast\\ast} & 4.27\\pm0.17^{\\ast\\ast} & \\text{OmniAudio} & 4.11\\pm0.15^{\\ast} & 4.16\\pm0.18^{\\ast\\ast} \\\\
> \\text{Ours} & 4.32\\pm0.15 & 4.44\\pm0.20 & \\text{Ours} & 4.31\\pm0.18 & 4.43\\pm0.22
> \\end{array}
> \\]
> In addition, we analyze inter-rater agreement. For each sample, we consider the ratings to be consistent if at least 3 out of the 5 raters assign the same score. Under this criterion, the agreement rates for MOS-SQ and MOS-AF are 68.75% and 65.42%(video-to-FOA), and 71.25% and 74.17%(text-to-FOA).
>
> Q5: Details of hyperparameters
>
> We hereby describe the corresponding parameters and their influence on latency:
> patch size: 4 latent frames, temporal stride: 4, causal context window: 2 patches(8 latent frames),NFE for LocDiT: 20.
>
> The 0.21s first-frame latency consists of 0.03s LM inference, 0.14s LocDiT denoising, and 0.04s video encoding and audio decoding. The iterative denoising nature of DiT makes this stage the dominant contributor, making NFE the key factor. Larger patch size increases both LM inference and denoising time, while a larger causal context window raises per-step latency. Thus, historical information(controlled by the causal context window) should be limited to minimize latency.
>
> Limitations: Due to space constraints, the OOD testing is presented in Reviewer mC6x's Q3.
>
> [1] Hu, Jinbo, et al. "PSELDNets: Pre-trained neural networks on a large-scale synthetic dataset for sound event localization and detection." IEEE Transactions on Audio, Speech and Language Processing (2025).

---

> > ### Author Rebuttal · Reviewer_1kt9 · 2026-04-03
> >
> > The rebuttal addresses my main concerns, so I increase my score to 4.

---

### Official Review · Reviewer_mC6x · 2026-03-11

**Soundness:** 3
**Presentation:** 4
**Significance:** 3
**Originality:** 3
**Overall Recommendation:** 4
**Confidence:** 4

**Summary:**

The paper proposes S3Audio, a streaming framework for generating high-fidelity spatial audio from panoramic videos and text prompts. It introduces a causal autoregressive diffusion transformer for streaming audio generation, along with SVAC contrastive learning and ODPO optimization to improve video–audio alignment and spatial perception. To address the lack of spatial audio data, the authors also develop an automated pipeline to generate spatial captions for training.

**Compliance With Llm Reviewing Policy:**

Affirmed.

**Key Questions For Authors:**

**Q1. Joint video+text performance**
 How does the model perform when **joint video+text inputs** are provided? Could the authors provide a comparison?

**Q2. Ablation on input modalities**
 Can the authors provide **ablation results on different input modalities** on the test set to clarify the contribution of each input type?

**Q3. Out-of-distribution evaluation**
 Have the authors evaluated the model on **out-of-distribution datasets**, such as **OmniAudio**, to assess generalization beyond the collected datasets?

**Q4. ViSAGe test configuration**
 How is **ViSAGe** configured during testing, given it requires FoV video input?

**Q5. OmniAudio retraining for text-to-audio**
 For generating spatial audio from text, was the **OmniAudio model** retrained on relevant data, or used directly?

**Limitations:**

Yes

**Strengths And Weaknesses:**

**Strengths**:

1. S3Audio introduces a causal autoregressive diffusion transformer architecture and effectively combines the SVAC strategy with multi-objective ODPO to achieve strong spatial perception from multimodal inputs, producing audio that aligns well with human preferences in aesthetics, semantics, and spatial understanding.
2. The paper builds the S3Audio corpus with an automated pipeline to scale high-quality spatial audio data and utilizes non-FOA audio to support the curriculum learning strategy during model training.

**Weaknesses**:

1. The model is trained using videos and optional text, and the paper evaluates these two input types separately. However, no comparison is provided for joint video+text inputs.
2. The authors should provide ablation results on different input modalities evaluated on the test set to show the impact of each input type.
3. Most experiments are conducted on the authors’ own collected datasets. Although baselines are trained on the same data for fairness, it would be valuable to also evaluate the model on an out-of-distribution dataset, such as OmniAudio, to assess generalization.
4. It is unclear how ViSAGe, which requires FoV video input, is configured during testing in Section 4.1. Additionally, for the task of generating spatial audio from text descriptions, did the authors retrain the OmniAudio model on the relevant data, or use it as-is? Clarification on these points is needed.

---

> ### Author Rebuttal · Authors · 2026-03-31
>
> Q1&Q2: Joint video+text performance and Ablation on input modalities
>
> Following the reviewer’s suggestion, we additionally evaluate joint video+text conditioning by providing both the panoramic video and the caption as inputs to the model. The results are shown below.
> Note that this evaluation is conducted on the caption test set, which is a subset of the video test set.
>
> \\[
> \\begin{array}{lccccc}
> \\text{Method} & \\text{FD} & \\text{KL} & \\Delta \\mathrm{abs}\\theta \\downarrow & \\Delta \\mathrm{abs}\\phi \\downarrow & \\Delta \\mathrm{angular} \\downarrow \\\\
> \\text{video} & 120.28 & 1.36 & 1.14 & 0.40 & 1.03 \\\\
> \\text{caption} & 142.80 & 1.43 & - & - & - \\\\
> \\text{video{+}caption} & 118.31 & 1.38 & 1.09 & 0.38 & 0.96
> \\end{array}
> \\]
> As can be seen, when joint video+text inputs are provided, the model achieves better generation quality than using either video-only or text-only input. In particular, compared with the video-only setting, adding captions further improves the spatial metrics . This suggests that captions provide additional and more explicit spatial cues, while joint video+text conditioning offers richer control signals for the model, leading to improved spatial consistency and overall generation quality.
>
> Q3: Out-of-distribution evaluation
>
> Our training set is composed of Sphere360 (the dataset introduced in OmniAudio), YT-Ambigen, and our newly collected dataset. The test set is constructed from the test splits of both Sphere360 and YT-Ambigen. In addition, to further evaluate the model’s generalization ability, we also conduct evaluation on the YT360-Test set. The results are shown below:
>
> \\[
> \\begin{array}{lcccccc}
> \\text{Method} & \\text{Params} & \\text{FD} & \\text{KL} & \\Delta_{\\mathrm{abs}}\\theta \\downarrow & \\Delta_{\\mathrm{abs}}\\phi \\downarrow & \\Delta_{\\mathrm{angular}} \\downarrow \\\\
> \\text{MMAudio+AS} & 1.03\\text{B} & 276.84 & 2.64 & - & - & - \\\\
> \\text{Diff-Foley+AS} & 0.94\\text{B} & 283.27 & 3.14 & - & - & - \\\\
> \\text{ViSAGe} & 0.36\\text{B} & 266.91 & 2.97 & 1.82 & 0.75 & 1.86 \\\\
> \\text{OmniAudio} & 1.22\\text{B} & 186.42 & 2.17 & 1.42 & 0.55 & 1.44 \\\\
> \\text{Ours} & 1.09\\text{B} & 145.83 & 1.60 & 1.24 & 0.45 & 1.13
> \\end{array}
> \\]
> These results show that our method achieves the best performance across both quality metrics and spatial accuracy metrics on YT360-Test, demonstrating stronger out-of-distribution generalization than baselines.
>
> Q4:  ViSAGe test configuration
>
> We follow the data processing pipeline of ViSAGe to extract FoV videos and energy maps from the training set, which are then used to train the ViSAGe baseline.
>
> Q5: OmniAudio retraining for text-to-audio
>
> For the text-to-spatial audio generation task, we do not directly use the original OmniAudio model. Instead, we retrain OmniAudio on our constructed caption dataset. Specifically, we replace the visual input with learnable null tokens, and encode the captions using a text encoder (Qwen3-0.6B). The resulting text features are then projected through a linear layer and incorporated into the global conditioning signal.

---

> > ### Author Rebuttal · Reviewer_mC6x · 2026-04-04
> >
> > Thank you for the rebuttal. I will keep my score unchanged.

---

### Decision · Program_Chairs · 2026-04-30

**Decision:**

Accept (regular)

**Comment:**

This paper introduces S3Audio, a streaming framework for high-fidelity spatial audio generation from panoramic videos and text prompts. The architecture utilizes an autoregressive diffusion transformer to decouple global semantic planning from local acoustic rendering.

All reviewers recommended acceptance (4 Weak Accepts). Reviewers were impressed by the practical streaming architecture, the impressive latency (0.21s first-chunk), and the contrastive learning (SVAC) strategy that forces the model to learn spatial awareness. Reviewers initially questioned the risk of reward hacking in the spatial metric, the lack of joint video-text evaluation, and the motivation for using DPO over GRPO.

During the rebuttal phase, the authors successfully addressed these critical issues. They used an independent spatial metric (PSELDNets) to validate spatial accuracy, evaluated baselines fairly using a trained predictor, demonstrated strong out-of-distribution performance, and justified their use of DPO for stable post-training. The reviewers found these additions convincing and acknowledged that their concerns were fully resolved.

The paper is technically solid, well-motivated, and advances the field of immersive media generation. Therefore, we recommend acceptance. Please ensure that the final camera-ready version incorporates all the changes promised during the rebuttal phase.